# *Untangle*:
# CRITIQUING DISENTANGLED RECOMMENDATIONS

## ABSTRACT

The core principle behind most collaborative filtering methods is to embed users and items in latent spaces, where individual dimensions are learned independently of any particular item attributes. It is thus difficult for users to control their recommendations based on particular aspects (critiquing). In this work, we propose Untangle: a recommendation model that gives users control over the recommendation list with respect to specific item attributes, (e.g.:less violent, funnier movies) that have a causal relationship in user preferences. Untangle uses a refined training procedure by training (i) a (partially) supervised $\beta$-VAE that disentangles the item representations and (ii) a second phase which optimized to generate recommendations for users. Untangle gives control on critiquing recommendations based on users preferences, without sacrificing on recommendation accuracy. Moreover only a tiny fraction of labeled items is needed to create disentangled preference representations over attributes.

## 1 INTRODUCTION

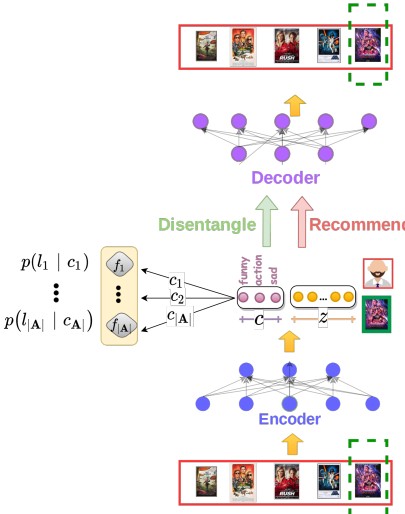

Figure 1: *Untangle* model is trained in two phases: Disentangling phase: Input to encoder is a one hot representation of an item (green dotted line). Obtained representation is disentangled across **A** attributes. Recommendation phase: Input to encoder is the items user interacted with (solid red line) and recommends new items.

User and item representations form the basis of typical collaborative filtering recommendation models. These representations can be learned through various techniques such as Matrix Factorization (1; 2), or are constructed dynamically during inference e.g. the hidden state of RNN's in session-based recommendations (3; 4).

As most standard recommendation models solely aim at increasing the performance of the system, no special care is taken to ensure interpretability of the user and item representations. These representations do not explicitly encode user preferences over item attributes. Hence, they cannot be easily used by users to change a.k.a. critique (5) the recommendations. For instance, a user in a recipe recommendation system cannot ask for recommendations for a set of less spicy recipes, as the spiciness is not explicitly encoded in the latent space. Moreover the explainability of the recommendations that are provided by such systems is very limited.

In this work, we enrich a state-of-the-art recommendation model to explicitly encode preferences over item attributes in the user latent space while simultaneously optimizing for recommendation's performance. Our work is

motivated by disentangled representations in other domains, e.g., manipulating generative models of images with specific characteristics (6) or text with certain attributes (7). Variational Autoencoders (VAEs), particularly $\beta$-VAE's (8) (which we adapt here), are generally used to learn these disentangled representations. Intuitively, they optimize embeddings to capture meaningful aspects of users and items independently. Consequently, such embeddings will be more usable for *critiquing*.

There are two types of disentangling $\beta$-VAEs: *unsupervised* and *supervised*. In the former, the representations are disentangled to explanatory factors of variation in an unsupervised manner, i.e., without assuming additional information on the existence (or not) of specific aspects. Used in the original $\beta$-VAE (8) approach, a lack of supervision often results in inconsistency and instability in disentangled representations (9). In contrast, in supervised disentangling, a small subset of data is assumed to have side-information (i.e. a label or a tag). This small subset is then used to disentangle into meaningful factors (10; 9). As critiquing requires user control using familiar terms/attributes, we incorporate supervised disentanglement in a $\beta$-VAE architecture in this work.

To achieve the explicit encoding of preferences over item attributes in embedding space we refine the training strategy of the *untangle* model. We essentially train in two phases: i) *Disentangling phase*: We explicitly disentangle item representations, using *very few* supervised labels. ii) *Recommendation phase*: We encode the user, using the bag-of-words representation of the items interacted, and then generate the list of recommended items. *Untangle* gives fine-grained control over the recommendations across various item attributes, as compared to the baseline. We achieve this with a tiny fraction of attribute labels over items, and moreover achieve comparable recommendation performance compared to state-of-the-art baselines.

## 2 Related Work

Deep learning based Autoencoder architectures are routinely used in collaborative filtering and recommendation models (11; 12; 13). In particular (11; 12) adopt denoising autoencoder architectures, whereas (13) uses variational autoencoders. The internal (hidden) representations generated by the encoders in these models are not interpretable and hence cannot be used for critiquing or explanations in recommendations.

Recent work on Variational Autoencoders across domains have focused on the task of generating disentangled representations. One of the first approaches used to that end was $\beta$-VAE (8; 14; 15), which essentially enforced a stronger (multiplying that term with $\beta > 1$) KL divergence constraint on the VAE objective. Such representations are more controllable and interpretable as compared to VAEs.

One of the drawbacks of $\beta$-VAE is that the disentanglement of the factors cannot be controlled and that they are relatively unstable and not easy to reproduce particularly when the factors of variance are subtle (9; 8; 14; 16; 17). This has motivated methods that explicitly supervise the disentangling (10), that rely either on selecting a good set of disentangling using multiple runs and the label information (18), or by adding a supervised loss function in the $\beta$-VAE objective function (10). As supervised disentangling methods are better in explainability and could provide control over desired attributes, we motivate our model from (19) for better critiquing in VAE based recommendation systems.

In recommender systems similar methods to utilize side information, have also been used recently to allow for models that enable critiquing of recommendations. These models allow users to tune the recommendations across some provided attributes/dimensions. Notable examples are (20; 21), where the models are augmented with a classifier of the features over which to control the recommendation. Adjusting the features at the output of the classifier modifies the internal hidden state of the model and leads to recommendations that exhibit or not the requested attribute. Note that this method of critiquing is quite different to our approach which allows for a gradual adjustment of the attributes. Moreover the models in (20; 21) require a fully labeled dataset with respect to the attributes while our approach only requires a small fraction of labeled data.

Unsupervised disentanglement was also recently used to identify and potentially use factors of variation from purely collaborative data i.e., data generated by user interactions with items (22) note though that this method focus was mainly on performance of the recommendations and that it does not allow for seamless critiquing as it is not clear what aspect of the data get disentangled.

## 3 Untangle

The aim of the *untangle* model is to obtaining controllable user (and item) representations for better critiquing along with optimizing for recommendation performance. To this end, we incorporate a simple supervised disentanglement technique to disentangle across item attributes/characteristics over which we want to provide explicit control to the users.

We index users with $u \in \{1, \dots, n\}$, and items with $i \in \{1, \dots, m\}$. $X^{n \times m}$ is a matrix of user-item interactions ($x_{ui} = 1$ if user $u$ interacted with item $i$, and 0 otherwise). A subset of items are assumed to have binary labels for attributes $\mathbf{A}$.

Our model is a modified $\beta$-VAE architecture, with a feed forward network based encoder and decoder. In Figure 1, user $u$ is represented by $[\mathbf{z} : \mathbf{c}]$. Note that : stands for concatenation, the $\mathbf{z}$ part of the representation is non-interpretable by default while on the $\mathbf{c}$ part of the representation we map (through a refined learning step) the representation of the attributes of the items over which we would like the user to have control. Each dimension in $\mathbf{c}$ is mapped to only one attribute $a$. Across the paper, we refer the dimension associated with the attribute $a$, as $c_a$. The user representation is sampled from the distribution parameterized by the encoder ($q_\phi$): $q_\phi(x_{u*}) = \mathcal{N}(\mu_\phi(x_{u*}), diag(\sigma_\phi(x_{u*}))$. The input to the encoder is the bag of words representation of the items $u$ interacted with, i.e. the $u^{th}$ row of matrix $X$, $x_{u*}$. The decoder generates the probability distribution given user representation $[\mathbf{z} : \mathbf{c}]$, $\pi(z) \propto exp(f_\phi^{dec}([\mathbf{z} : \mathbf{c}]))$, over the $m$ items. The likelihood function used in recommender system settings (3; 23; 24; 25) is typically the multinomial likelihood:

$$p_\theta(x_u|[\mathbf{z} : \mathbf{c}]) = \sum_i x_{ui} \log \pi_i([\mathbf{z} : \mathbf{c}]))$$

### 3.1 LEARNING

Training is conducted in two phases: Recommendation and Disentangle phase, as mentioned in Algorithm 1.

**Recommendation Phase** The objective in this phase is to optimize the encoder parameterized by ($\theta$), and decoder parameterized by ($\psi$) to generate personalized recommendations. We train our model with the following objective:

$$L(x_{u*}, \theta, \phi) \equiv \mathbb{E}_{q_\theta([\mathbf{z}:\mathbf{c}]|x_{u*})}[log p_\theta(x_{u*}|[\mathbf{z} : \mathbf{c}])] - \beta KL(q_\phi([\mathbf{z} : \mathbf{c}]|x_{u*})|p([\mathbf{z} : \mathbf{c}])) \quad (1)$$

Intuitively, this is the negative reconstruction error minus the Kullback-Leibler divergence enforcing the posterior distribution of $\mathbf{z}$ to be close to the Gaussian distribution (prior) $p(\mathbf{z})$.

The KL divergence in $\beta$-VAE is computed between the representation sampled from the encoder and the normal distribution $p(\mathbf{z}) = \mathcal{N}(0, I_d)$. The diagonal co-variance matrix enforces a degree of independence among the individual factors of the representation. Consequently, increasing the weight of the KL divergence term with $\beta > 1$ boosts the feature independence criteria, leading to disentangled representation. This ensures that even in the recommendation phase, the learnt user representations are nudged towards disentanglement.

**Disentanglement Phase** Since the attribute information is commonly available across the items. In this phase, we first obtain the item representation in the user latent space (as depicted in the highlighted green box in Figure 1). We pass the one hot encoding of an item, and obtain its representation in the latent user space. We then disentangle the obtained representation using the following objective:

$$L(\mathbf{1}_i, \theta, \phi) \equiv \mathbb{E}_{q_\theta([\mathbf{z}:\mathbf{c}]|\mathbf{1}_i)}[log p_\theta(\mathbf{1}_i|[\mathbf{z} : \mathbf{c}])] \quad (2)$$
$$- \beta KL(q_\phi([\mathbf{z} : \mathbf{c}]|\mathbf{1}_i)|p([\mathbf{z} : \mathbf{c}])) + \gamma \mathbb{E}_{q_\theta(c|\mathbf{1}_i)} l(q_\phi(\mathbf{c}|\mathbf{1}_i), \mathbf{a})$$

---

**Algorithm 1:** *Untangle:* Training

**Data:** $X \in R^{n \times m}$ containing user-item interactions, with a subset of items having labels for $\mathbf{A}$ attributes

1   initialize model params.: Encoder($\phi$), Decoder($\theta$) ;
2   **do**
3     **if** *is_disentangle* **then**
       // Disentangle representations
4        $\mathbf{1}_i \leftarrow$ random mini batch from set of items that are labelled with $\mathbf{A}$ set.
5        $[\mathbf{z} : \mathbf{c}] \leftarrow$ sample from $\mathcal{N}(\mu_\phi(\mathbf{1}_i), diag(\sigma_\phi(\mathbf{1}_i))$
6        $\tilde{x}_{i*} \leftarrow$ Decoder($[\mathbf{z} : \mathbf{c}]$)
7        compute gradients $\nabla L_\phi, \nabla L_\theta$ using Objective 2
8        $\phi \leftarrow \phi + \nabla L_\phi$
9        $\theta \leftarrow \theta + \nabla L_\theta$
10     **end**
11     **if** *is_recommend* **then**
       // Recommend items
12        $\mathbf{x}_{u*} \leftarrow$ random mini-batch from dataset
13        $[\mathbf{z} : \mathbf{c}] \leftarrow$ sample from $\mathcal{N}(\mu_\phi(x_{u*}), diag(\sigma_\phi(x_{u*}))$
14        $\tilde{x}_{u*} \leftarrow$ Decoder($[\mathbf{z} : \mathbf{c}]$)
15        compute gradients $\nabla L_\phi, \nabla L_\theta$ using Objective 1
16        $\phi \leftarrow \phi + \nabla L_\phi$
17        $\theta \leftarrow \theta + \nabla L_\theta$
18     **end**
19   **while** *model converges*;

---

As in (10), we modify the $\beta$-VAE objective (Objective 1) in to incorporate a classification loss over the factors $\mathbf{c}$, over which we disentangle. This loss penalizes discrepancies between the attribute label prediction for factor $c_a$ and the label $a$ of interest, nudging the disentanglement for each attribute to happen over the corresponding factor $c_a$.

## 4   Datasets

**Movielens Dataset:** We use the Movielens-1m and Movielens-20m datasets (26), which contain 1 million and 20 million user-movie interactions, respectively. For the latter, we filter out movies with fewer than 5 ratings and users who rated $\leq 10$ movies. We utilize the relevance scores given in the Movielens dataset for 10,381 movies across 1,000 different tags to select attributes for disentangling. E.g., *Mission Impossible* movie has high relevance (0.79) for the *action* tag. We take the top 100 tags, based on the mean relevance score across all movies. Among these 100 tags, some tag pairs, like (*funny*, and *very funny*), are by definition entangled. Therefore, to identify distinct tags, we cluster these 100 tags ($\in \mathcal{R}^{10381}$ movies) into 20 clusters using K-Means clustering. Finally, we select a subset from these 20 clusters, as given in Table 1 for disentangling. We assign the new-clustered tag (as given in Table 1, Column 1) if the average-relevance score (the mean of relevance scores for tags present in the corresponding cluster) is higher than 0.5.

**Goodreads Dataset:** The GoodReads dataset (27) contains user-book interactions for different genres. We use the Children and Comics genres to evaluate our model. We filter out items rated $\leq 5$ and users who rated $\leq 10$ books. The final statistics are given in Appendix A. We extract the tags for disentangling from the user-generated shelf names, e.g., *historical-fiction, to-read.* We retrieve the top 100 shelf names. Some tags (like "books-i-have") are not useful to revise recommendations. Therefore, we only consider item attributes that all the authors consider *informative* for critiquing recommendations. We select a subset for disentangling from this set, as it still contains correlated attributes like *historical-fiction, fiction.* We select attributes with the corresponding number of books where the attribute was present {horror:1080, humor:9318, mystery:3589, and romance:1399} and {adventure:8162, horror:5518, humor:8314, mystery:5194, romance:7508, sci-fi:7928}, for Goodreads-(Children and Comics) respectively.

| Cluster Label | Tagged movies | Tags included in cluster |
|---|---|---|
| action | 1,167 | action, fight-scenes, special-effects |
| funny | 1,219 | comedy, funny, goofy, very funny |
| romantic | 975 | destiny, feel-good, love story, romantic |
| sad | 1,488 | bleak, intimate, loneliness, melancholic, reflective, sad |
| suspense | 1,070 | betrayal, murder, secrets, suspense, tense, twist-and-turns |
| violence | 1,297 | brutality, cult classic, vengeance, violence, violent |

Table 1: Each cluster was manually assigned a human-readable label. Some of the tags present in each cluster are listed in column 3. Column 2 lists the number of movies that had high relevance score for tags in each cluster.

## 5 Evaluation Metrics

We evaluate *Untangle* on these criteria: i) quality of items recommended, ii) extent of disentanglement, iii) control/critiquing based on the disentangled representations.

**Ranking Based Metrics:** We evaluate the quality of items recommended using two ranking-based metrics: Recall@k and normalized discounted cumulative gain NDCG@k. The latter is rank sensitive, whereas Recall@k considers each relevant item in the top-k equally.

$$Recall@k := \frac{\sum_{i=1}^{k} \mathbb{I}[item[i] \in \mathcal{S}]}{min(k, |\mathcal{S}|)} \qquad DCG@k := \sum_{i=1}^{k} \frac{2^{\mathbb{I}[item[i] \in \mathcal{S}]} - 1}{\log(i+1)}$$

NDCG is normalized DCG by dividing it by the largest possible DCG@k.

**Disentanglement Metrics:** We use the *Disentanglement*, and *Completeness* metrics introduced in (28). *Disentanglement* measures the extent to which each dimension captures at most one attribute. E.g., if a dimension captures all attributes, the Disentanglement score will be 0. We compute *importance* $p_{aj}$ of $a^{th}$ attribute on $j^{th}$ dimension of $[\mathbf{z} : \mathbf{c}] \in \mathcal{R}^d$, with Gradient Boosted Trees as given in (9). Using the $p_{aj}$ scores, the disentanglement score is defined as:

$$H_{|\mathbf{A}|}(P_j) = -\sum_{a=0}^{|\mathbf{A}|-1} p_{aj} log_{|\mathbf{A}|} p_{aj}, \quad D_j = (1 - H_{|\mathbf{A}|}(P_j))$$

$$\mathbf{D} = \sum_{j=0}^{d-1} \rho_j D_j, \quad \rho_j = \frac{\sum_{a=0}^{|\mathbf{A}|-1} p_{aj}}{\sum_{j=0}^{d-1} \sum_{a=0}^{|\mathbf{A}|-1} p_{aj}}$$

We compute entropy $H_{|\mathbf{A}|}(P_j))$ for $j^{th}$ dimension. Disentanglement score for dimension $j$ is then $1 - entropy$. The final disentanglement score of the system is weighted average of $D_j$ across all the dimensions $d$, where $\rho_j$ the dimension's relative importance. *Completeness*: Measures the extent to which one attribute $a$ is encoded in a single dimension of $[\mathbf{z} : \mathbf{c}]$. For a latent representation of 16 dimensions and 2 attributes, if 8 dimensions encode attribute $a_1$ and the other 8 encode $a_2$, then the *Disentanglement* will be 1 but *Completeness* will be 0.25. Completeness is defined as:

$$H_d(P_a) = -\sum_{j=0}^{d-1} p_{aj} log_d p_{aj}, \quad C_a = (1 - H_d(P_a))$$

$$\mathbf{C} = \sum_{a=0}^{|\mathbf{A}|-1} \rho_a C_a, \quad \rho_a = \frac{\sum_{j=0}^{d-1} p_{aj}}{\sum_{a=0}^{|\mathbf{A}|-1} \sum_{j=0}^{d-1} p_{aj}}$$

**Controller Metric:** We propose a simple metric to quantify the extent of *control* disentangled dimension $c_a$ has on recommendations by critiquing attribute $a$. With supervised-disentanglement, the mapping between dimensions $\mathbf{c}$ in the latent representations, and the attributes across which we disentangled is known. The features in these dimensions in $\mathbf{c}$ allow the user to control/critique the respective attribute in the generated recommendations. For instance, *less violence* can be achieved by reducing the corresponding dimension value

| Dataset | Model | Recommendation Performance | | | Disentanglement Performance | | |
|---|---|---|---|---|---|---|---|
| | | *N@100* | *R@20* | *R@50* | *Disent.* | *Comp.* | *Controller Metric* |
| **ML-1m** | Multi-DAE | 0.38782 | 0.31636 | 0.43404 | 0.317 | 0.214 | 0.961 |
| | Multi-VAE | **0.39252** | **0.32515** | **0.44757** | 0.306 | 0.200 | 0.947 |
| | $\beta$-VAE | 0.38658 | 0.31216 | 0.43032 | 0.313 | 0.0211 | 0.924 |
| | *Untangle* | 0.37833 | 0.30079 | 0.42532 | **0.543** | **0.393** | **19.27** |
| **ML-20m** | Multi-DAE | 0.39738 | 0.37071 | 0.50847 | 0.265 | 0.182 | 0.88 |
| | Multi-VAE | 0.39827 | 0.37212 | 0.50946 | 0.246 | 0.167 | 3.53 |
| | $\beta$-VAE | 0.38724 | 0.35617 | 0.48976 | 0.211 | 0.142 | 3.27 |
| | *Untangle* | **0.40320** | **0.37367** | **0.51303** | **0.677** | **0.529** | **75.11** |
| **GR-Comics** | Multi-DAE | 0.42593 | 0.42602 | 0.52610 | 0.243 | 0.175 | 0.963 |
| | Multi-VAE | 0.45159 | 0.45697 | 0.55598 | 0.173 | 0.137 | 0.872 |
| | $\beta$-VAE | **0.44366** | **0.44949** | **0.55226** | 0.192 | 0.146 | 0.847 |
| | *Untangle* | 0.43597 | 0.43981 | 0.54218 | **0.733** | **0.536** | **73.41** |
| **GR-Children** | Multi-DAE | 0.40030 | 0.43240 | 0.56473 | 0.145 | 0.132 | 2.37 |
| | Multi-VAE | 0.40219 | 0.43057 | 0.56695 | 0.164 | 0.132 | 0.86 |
| | $\beta$-VAE | 0.40219 | 0.43057 | 0.56695 | 0.139 | 0.103 | 0.92 |
| | *Untangle* | **0.41255** | **0.44490** | **0.58473** | **0.517** | **0.574** | **14.37** |

Table 2: Recommendation and Disentanglement performance on Movielens-(1m,20m) and Goodreads-(Comics,Children) domain dataset on the corresponding test split.

(violence) in **c**. We evaluate this by probing if the items where the attribute is present $(\mathcal{S}_a)$ are ranked higher when the dimension value $c_a$ is increased by a factor of $g$ in the user representation. We extract the items recommended from the decoder $(\mathcal{I}_a(g))$, for the new user representation where *only* $c_a$ is multiplied $g \times c_a$. We compare $(\mathcal{I}_a(g))$ against $(\mathcal{S}_a)$ using any ranking-based metric described above. We further vary $g$ for a given range $[-G, G]$, and study if the ranking of $(\mathcal{S}_a)$ improves. The Controller-Metric is defined as follows:

$$\text{Controller\_Metric}(k, g) := \frac{|Recall@k(\mathcal{I}_a(G), \mathcal{S}_a) - Recall@k(\mathcal{I}_a(-G), \mathcal{S}_a)|}{Recall@k(\mathcal{I}_a(-G), \mathcal{S}_a)} \tag{3}$$

To compute the Controller-Metric for a system, we take the median across all the attributes disentangled in **c**. Note that the metric value depends on $k$ and the range chosen.

## 6 RESULTS AND DISCUSSIONS

**Recommendation and Disentanglement Performance** We train the *Untangle* model with the parameter settings mentioned in Appendix B. We compare *Untangle* with the MultiDAE, and MultiVAE models (13). We also compare our model with a stronger baseline for disentanglement $\beta$-VAE, which disentangles the representation in an unsupervised way. We present our results in Table 2. Note that supervised disentanglement for Table 2, has been trained with 300 (1%), 1030 (5%), 1500 (5%), 1550 (5%) labelled items for Movielens-(1m,20m) and Goodreads-(Children,Comics) respectively. We observe that our proposed model's performance on ranking-based metrics (Recall@k, and NDCG@k) is comparable to the baselines across all datasets. Thus we show that disentangling the latent representation does not impact the recommendation performance. We also quantify the disentanglement using the Disentanglement and Completeness metrics discussed in Section 5. We infer from Table 2 that the disentanglement achieved across all the mentioned strategies is significantly higher than the baselines. Disentangling with a tiny fraction of labeled items leads to a significant gain in disentanglement compared to $\beta$-VAE.

We evaluate the extent of the *controllability* of the disentangled representations. To this end, we compute the Controller Metric, which measures the control over the attribute dimension $c_a$ variation. We use the multiplicative range of $[-150, +150]$ to amplify $c_a$, and measure the ranking performance using recall@10 across this range. Note that the rest of the representation remains unchanged. We observe that we get significantly higher *controllability* for the *Untangle* model compared to the baseline approaches, especially for Movielens-20m and Goodreads-Comics dataset. By reducing $c_a$ we can diminish the existence of items with attribute $a$ from the recommendation list and by gradually increasing the magnitude of $c_a$ increase the presence of items with this attribute in the recommendation list up to saturation.

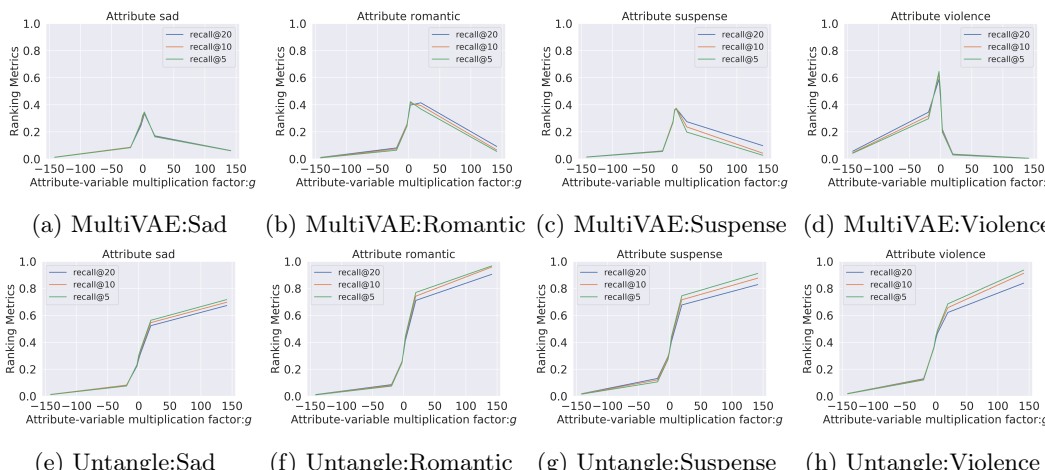

Figure 2: Control over recommendations when factor-value $c_a$, is adjusted by multiplicative factor $g \in [-150, 150]$. Recommendation lists are evaluated by recall@(5,10,20). Relevance is determined by the presence of attribute $a$ in the retrieved items. We compare Multi-VAE (top) with Untangle model (bottom) for sad, romantic, suspense and violence on ML-20m.

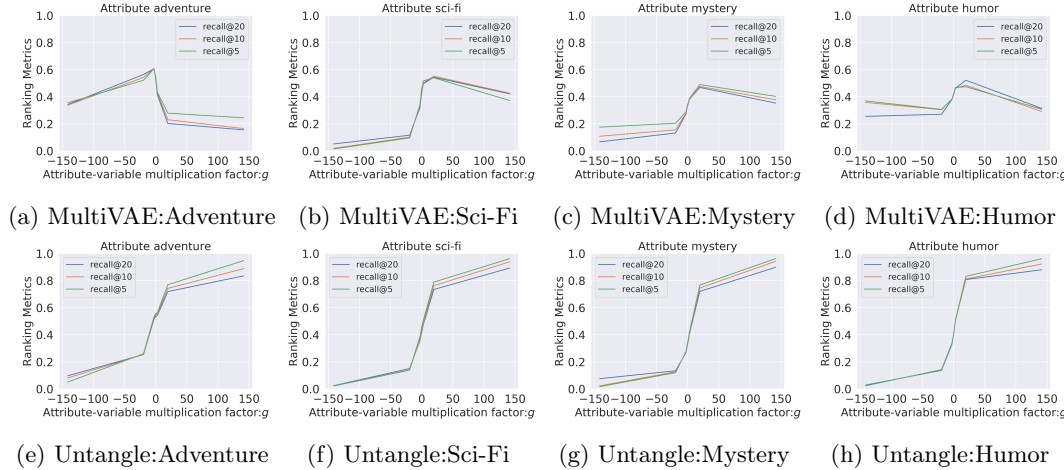

Figure 3: We compare Multi-VAE (top) with Untangle model (bottom) for adventure, sci-fi, mystery and humor attributes for Goodreads-Comics for the same analysis done in Figure 2.

**Critiquing Recommendations** The primary aim of our model is to obtain *controllable* representations for critiquing. With the Controller Metric, we quantify *controllability*, here we further analyze the incremental impact of changing the attribute dimension. In this analysis, we visualize the effect on the recommendations of the adjustment of the disentangled factor $c_a$ for each attribute $a$. We multiply the factor with $g$ in Figure 2 and Figure 3 for baseline model MultiVAE and *Untangle*. Note that for the baseline (MultiVAE), we adjust the dimension that has the highest feature importance score computed using Gradient Boosting Classifier for attribute $a$.

For the movies domain (Figure 2), we observe that for MultiVAE (row 1) the variation in $c_a$ has no clear correlation with the recommendation performance in terms of the presence or absence of items with this attribute. In contrast to MultiVAE, in the *Untangle* model, we consistently observe a significant and gradual variation across all the explicitly disentangled attributes **A**. Even for subtle attributes like suspense, we obtain a complete range of recall@10 from 0.0 to 1.0 We observe similar results for Goodreads comics dataset (Figure 3), where we again get gradual and significant change (approximately 1) across all the disentangled attributes.

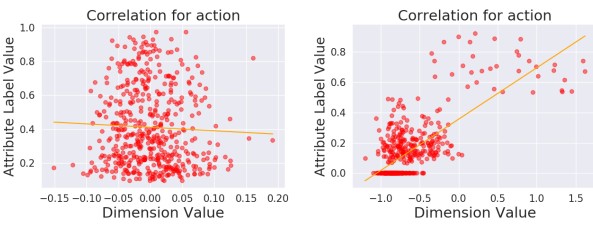

Figure 4: Correlation between learnt dimension value $c_a$ to the true relevance score across 500 movies for Movielens-20m

**Correlation between Relevance Scores and $c_a$ :** We observe that disentangling across item representations leads to a fine-grained control for critiquing. We further verify, if the achieved controllability is an outcome of high correlation between factor $c_a$, and the true relevance score across movies for attribute $a$ for Movielens-20m dataset. We randomly sample 500 movies, and obtain their latent representation from the encoder. In Figure 4, we plot the obtained $c_a$ value to and the true relevance score for attribute *action*. We can infer from the Figure 4 that the representations obtained from *Untangle* have a high Pearson correlation of 0.53 as compared to MultiVAE model (Pearson Correlation: -0.03). The graphs for other attributes/tags are presented in Appendix C.

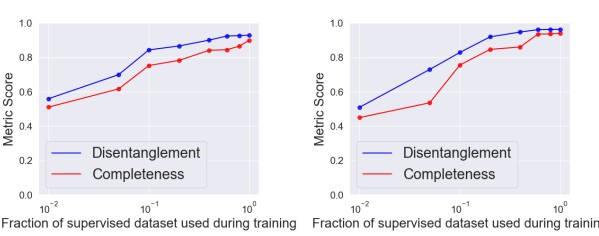

Figure 5: Variation in Disentanglement and Completeness metrics when model is trained with lesser labels for Movielens-20m and GoodReads-Comics.

**Fewer Labels for Disentanglement** One of the advantages of *Untangle* is that it disentangles with very few labels. We train *Untangle* with fewer labeled items. Each point in in Figure 5 is an average across 5 different runs with different random seeds. For Movielens-20m just 1% attribute labels yields a disentanglement score of 0.51, which gradually increases up to 0.92 when trained with all labels. For Goodreads-Comics, with 1% labelled books we are able to achieve 0.52 disentanglement, which gradually increases to 0.93 when the model is trained with all the labels. Note that even with 1% labelled items, the disentanglement and completeness scores obtained are significantly higher than $\beta$-VAE model:0.21 and 0.19 on Movielens-20m and Goodreads-Comics and respectively.

**Controllable Attributes** With the above analysis, we have established that *Untangle* leads to controllable representations. In this experiment, we identify if the controllability is restricted to the chosen set of attributes. Therefore, we apply *Untangle* to a larger set of tags for Movielens-20m dataset. We cluster all the 1181 tags present in the dataset, using K-Means clustering into 50 clusters. The clustering strategy is similar to the one mentioned in Section 4. We then evaluate the controllability for each of the clustered-tag, $b$. We explicitly encode the corresponding clustered-tag $b$ using *Untangle*, using 5% of labelled items. The controller metric score is obtained for each tag, across 5 runs. In each run, we sub-sample four clustered tags out of 40 to be disentangled along with the corresponding clustered tag $b$. This is done to model the impact of disentangling a given attribute alongside with other attributes present in the dataset. We identify that across 40 clustered-tags, we obtain a controller-metric score of $> 11.0$ for over 21 tags. Some of the attributes which do not have a higher controller-metric score includes:80s, crappy, philosophical, etc. These attributes are also unlikely to be critiqued by user. Some of the most controllable and least controllable tags have been listed in Appendix D.

## 7    CONCLUSION

*Untangle* archives the goals we set, it provides control and critiquing over the user recommendations over a set of predefined item attributes. It does so without sacrificing recommendation quality and only needs a small fraction of labeled items.

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

## A  DATASET STATISTICS

We have mentioned the number of interactions, users, and items for Movielens and Goodreads Dataset in Table 3.

| Dataset | Number of Interactions | Number of Users | Number of Items | Sparsity Rate |
|---|---|---|---|---|
| Movielens-1m | 1,000,209 | 6,040 | 3,706 | 4.468 % |
| Movielens-20m | 9,990,682 | 136, 677 | 20, 720 | 0.353 % |
| Goodreads-Children | 3,371,518 | 92,993 | 33,635 | 0.108 % |
| Goodreads-Comics | 2,705,538 | 57,405 | 32,541 | 0.145 % |

Table 3: Dataset statistics (after performing all filtering). The sparsity rate indicates the fraction of cells in the complete user-item matrix with a known value.

## B  IMPLEMENTATION DETAILS

We divide the set of users into train, validation and test splits. Validation and test splits consist of 10% of the users, across all datasets. For each user in the validation and test split, we use only 80% of the items rated by them to learn the user representation. The remaining 20% is used to evaluate the model's performance. This strategy is similar to the one used by (13). For all the experiments, user's latent representation is restricted to 32 dimensions. The encoder and decoder consists of two layers with $[600, 200]$ and $[200, 600]$ hidden units respectively, each with ReLu activation. We conduct hyper-parameter tuning to identify $\beta$ and $\gamma$ values from $[5, 10, 50]$ and $[5, 10, 50, 500]$ respectively. The threshold $M$ to identify movies where the attribute is present for Movielens-20m , and MovieLens-1m is taken as 0.5 and 0.4 respectively. All the models are run up to 50 epochs. We select the best model, based on its performance on validation dataset for both NDCG@100 and Disentanglement score. We select less than 5% of items for supervised $\beta$-VAE using stratified sampling.

## C  CORRELATION BETWEEN DIMENSION VALUE $c_a$ AND TRUE RELEVANCE SCORES ACROSS ITEMS

We compare the dimension value $c_a$ associated with an attribut $a$, to the true relevance scores present in the Movielens-20m dataset. We show in Figure 6 that across all the tags, the correlation is consistently higher for Untangle, when compared to MultiVAE.

## D  CONTROLLABLE ATTRIBUTES

Using *Untangle*, we identify the clustered-tags, which are more controllable for revising user recommendations. We have listed some of the most controllable and least controllable tags in Table 4. We also list the absolute recall difference obtained across each cluster.

| Recall Difference | Tags in the cluster: |
|---|---|
| **10 Most Controllable Attributes** | |
| 0.75933 | action packed, adventure, big budget, cool, dynamic cgi action, exciting, fast paced, fighting, franchise, plot holes, series |
| 0.75924 | atmospheric, bleak, character study, downbeat, forceful, grim, masterpiece, movielens top pick, powerful ending, tense, visceral |
| 0.75461 | corruption, intense, murder, police investigation, secrets, suspense, suspenseful, thriller, twists & turns |
| 0.75246 | beautiful scenery, betrayal, childhood, earnest, excellent, excellent script, exceptional acting, friendship, good acting, great movie, honest, idealism, justice, light, moral ambiguity, original plot, oscar, oscar winner, sacrifice, unlikely friendships, very good, witty |
| 0.72529 | classic, cult classic, gunfight, highly quotable, quotable |
| 0.72285 | comedy, funny, hilarious, humorous, very funny |
| 0.7144 | afi 100 (movie quotes), oscar (best actor), oscar (best cinematography), oscar (best picture), oscar (best supporting actor) |
| 0.70965 | adapted from:book, based on a book, based on book, books |
| 0.61973 | future, futuristic, sci fi, sci-fi, science fiction, scifi, special effects, technology |
| 0.59895 | goofy, silly, silly fun |
| **10 Least Controllable Attributes** | |
| 0.24986 | erotic, sex, sexual, sexuality |
| 0.24014 | adolescence, bullying, coming of age, coming-of-age, high school, school, teacher, teen, teen movie, teenager, teenagers, teens |
| 0.23056 | anti-semitism, anti-war, best war films, bombs, civil war, fascism, genocide, german, germany, historical, history, holocaust, jewish, jews, military, nazi, nazis, poland, russian, war, war movie, wartime, world war i, world war ii, wwii |
| 0.17843 | broadway, dance, dancing, great music, hip hop, lyrical, music, music business, musical, musicians, rock and roll |
| 0.17675 | adapted from:comic, based on a comic, based on comic, comic, comic book, comics, graphic novel, mutants, super hero, super-hero, superhero, superheroes, vigilante |
| 0.1112 | business, capitalism, controversial, documentary, factual, freedom, islam, journalism, oil, political, politics, propaganda, revolution, us history, world politics |
| 0.08376 | 1970s, anti-hero, awesome soundtrack, california, crime, drugs, gangs, good music, great soundtrack, gritty, nostalgic, small town |
| 0.06328 | assassination, black comedy, brainwashing, censorship, cynical, distopia, fighting the system, guilt, hotel, identity, intellectual, intelligent, ironic, manipulation, morality, off-beat comedy, oscar (best writing - screenplay written directly for the screen), paranoid, philosophical, philosophy, surveillance, thought-provoking |
| 0.0432 | mentor, original |
| 0.00691 | 80s, awful, bad, bad acting, bad cgi, boring, camp, campy, cheesy, disaster, dumb, dumb but funny, horrible, idiotic, lame, mad scientist, nudity (topless), remake, ridiculous, stupid, stupid as hell, stupidity |

Table 4: Most controllable and least controllable tags obtained from Untangle

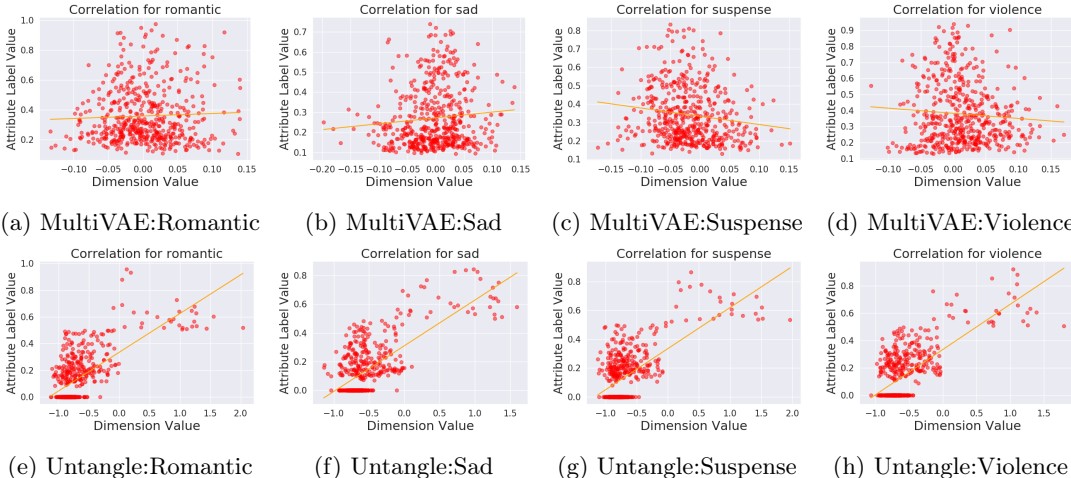

Figure 6: We compare Multi-VAE (top) with Untangle model (bottom) for the correlation between factor $c_a$ and true relevance scores.

