# OpenReview forum: "Untangle: Critiquing Disentangled Recommendations"
_ICLR.cc/2021/Conference — Reject_

### Official Review · AnonReviewer1 · 2020-10-27
**OK submission but lack of novelty and thorough experiments**

**Rating:** 5
**Confidence:** 5

**Review:**

The proposed approach leverages the beta-vae property that disentangles latent representation to do the critiquing operation on the recommendation system. A portion of latent representations dimensions of beta-VAE is further regularized by optimizing an additional objective that minimizes the loss of reconstructing some attributes of items, which enforce the particular latent dimension to represent a certain property of an item.

In the novelty perspective, the paper is based on disentangled beta-vae, which is well-established work. While applying the approach to the critiquing-based recommendation system is interesting, it does not hide the weakness of the novelty of this work (technically no much difference).

In terms of clearness, the paper is fairly written with reasonable description flow. However, some important information is missing. For example, in Equation 2, the author should emphasize the additional objective component by clearly state what $l$ function is. The objective function seems not correct as the expectation is on a single $c$ variable, but the loss is on a vector of $c$. I am not sure if this is a typo or intentional changes.

The author claims that the approach is distinguishable from the existing work [20, 21] since the existing work relies on data with attribute labels. However, when the paper introduces Equation 2, such a claim is already violated. Similarly, the attributes table (Table 1) also indicates such an effort of fetching attribute labels, which is no much different from Table 1 of [20]. I suggest the authors rephrase the related work section to clarify the similarity or difference. Meanwhile, I note that there is a paper that also makes a critiquing-based recommendation on VAE architecture. It should worth cite.

The experiment part does not fully reflect the paper's main claimed contribution in terms of critique. As the paper title stated, this paper leverage untangle to do the critiquing operation on recommender systems. While the author stated several existing works in the related work section, there is no comparison in the experiment but many the controllable attributes analysis (multiple figures deliver the exact same information.) Especially, such controllable attribute analysis has been well stated in the supervised beta VAE literature. The critiquing-based recommendation has a long history, and many classic approaches could be compared.

Recommendation performance in Table 2 is not quite satisfactory as the proposed model's performance performs suboptimally on 2 out of 4 datasets. While the controller metric is far higher than the others, it is an unfair comparison as others baselines never optimize (or intended to optimize) such objective.

In terms of metric, the paper does not evaluate the proposed model's critiquing performance with proper metric (this is not a controller metric). MAP deduction (or so-called F-MAP) in [20] is a good example to evaluate.


Some related papers:
[1] Luo, Kai, et al. "Deep Critiquing for VAE-Based Recommender Systems." Proceedings of the 43rd International ACM SIGIR Conference on Research and Development in Information Retrieval. 2020.

---

### Official Review · AnonReviewer3 · 2020-10-28
**An interesting paper but below the bar**

**Rating:** 4
**Confidence:** 3

**Review:**

Summary:
The paper proposes a framework to learn disentangled representations for collaborative filtering systems. To model the user-item interactions, the authors adopt the likelihood model proposed by \beta-Multi-VAE. The auxiliary task of predicting item labels is considered to increase the disentanglement and the ability to control the recommendations. Practical performance and the properties of disentanglement are demonstrated in experiments on real data sets.

Pros:
1. The paper focuses on a novel and important area for the recommender system. Learning disentangled representation might help obtain a model that is useful yet interpretable and controllable.
2. Consistent and superior disentanglement performances on all datasets.
3. The choice of experimentation metric is complete and the results are presented coherently.

Cons:

1.  While being neglected by most literature. Using disentangled representation for recommendation is not entirely new. Therefore this paper needs a stronger baseline (e.g. [https://papers.nips.cc/paper/7174-learning-disentangled-representations-with-semi-supervised-deep-generative-models]) for disentanglement learning models besides the \beta-Multi-VAE.

2. The idea of utilizing external knowledge or labels in the recommendation is not new. And it’s not uncommon for such models to tolerate missing contextual attributes.  [https://dl.acm.org/doi/10.1145/3097983.3098094] How does the proposed model compare with these baselines?

3. The selection of cut-off for ranking metrics is not very consistent. The superiority of the VAE models is sensitive to this choice as pointed in [https://dl.acm.org/doi/10.1145/3298689.3347058]

4. The setting of controller metric experimentation seems to be trivial or unfair. The multiplier is applied to the dimension c_a, which is only defined for models trained with attribute labels.

5. The presentation of the model is confusing. 1) What is 1_i in equation (2)? is it a bag of word representation of randomly sampled items? If it is, does it make sense to ask the model to approximate this random sample? And if it is not, then why is it used in the multinomial likelihood? 2) How does the model train? Does it train each phase at each iteration? Or it firstly trains the disentangle phase then the recommendation phase?

---

### Official Review · AnonReviewer2 · 2020-10-28
**An interesting work but may need some improvements**

**Rating:** 4
**Confidence:** 4

**Review:**

##########################################################################

Summary:

This paper proposed a method (Untangle) which allows users to control over the recommendations with respect to specific item
attributes. The proposed method leverages a supervised $\beta$-VAE to disentangle item representations, and an additional step to generate recommendations. The proposed method is compared with some unsupervised baseline methods through experiments on two real-world datasets.

##########################################################################

Strength:

- Interesting topic and practical in real-world application
- Relatively extensive experiments are conducted

##########################################################################

Weakness:

- Technical contribution/depth of this work is a bit limited
- Execution of the experiments and evaluation may need some improvements
- The overall presentation can be further improved

##########################################################################

Detailed Comments:

This work proposed an interesting angle to disentangle item representations and I personally like the pitch of recommendation interpretability and give control back to users. However the overall execution of this work may need some improvements, and I personally feel ICLR may not be the most appropriate venue for it, thus I tend to vote for a *rejection*.

- With the current framing of the paper, I feel the technical depth of the proposed method (i.e. the two-phase supervised $\beta$-VAE ) is a bit limited, especially considering the ICLR community. However I do like the interpretability and customization angle, as well as the efforts of exploring corresponding evaluation metrics. With that being said, a possibly better direction for improvement is re-framing and re-organizing the work to address the these pain points directly, rather than promote the algorithm standalone. The evaluation could also involve qualitative studies and get feedback from users directly. Regarding this I don't feel ICLR is an appropriate venue and recommendation/application-focused venue might be a better fit (e.g., RecSys/TheWebConf/WSDM/etc.).
- Another major concern for the quantitive evaluation is, the proposed method is only compared to unsupervised baselines. One may argue that it may not be a fair comparison as the supervised reconstruction method leverages additional information, thus additional baselines with the same type of information may need to be considered.


Minor Concerns:

- There are some typos and grammar mistakes in the draft and the overall presentation is a bit mouthful (not notably affect my reading but worth double checking and improve in the next iteration)
- The citation format in this draft looks informal (and different from other submissions in my queue) - please double check the latex template of the submission

---

### Official Review · AnonReviewer4 · 2020-10-28
**Good motivation, weak technical contribution, and weak evaluation**

**Rating:** 5
**Confidence:** 4

**Review:**

The submission aims to develop a disentangled recommender that explicitly grants the users the ability to control a chosen aspect of the recommendation list. This is achieved by (1) learning disentangled user/item representations via beta-VAE and (2) aligning text tags with certain dimensions via an auxiliary loss (the term after gamma in Eq 2). Empirical results demonstrate that this supervised approach outperforms the unsupervised baselines in terms of disentanglement and controllability, e.g., it can explicitly control the recommender to recommend a list of romantic movies.

Strengths:

S1: Good motivation. Controllability is potentially valuable for recommender systems.

S2: The proposed technique makes sense and is easy to implement.

S3: The writing is clear and easy to follow.

Weaknesses:

W1: The abstract states that the submission is interested in how to recommend *less* violent movies, *funnier* movies, etc. However, the empirical results do not fully support this. In fact, what this submission achieves is how to recommend a movie that is violent or not violent (i.e., whether it is of the category action movies or not) (binary controllability). There is no evidence that the proposed technique can tell the difference between “less violent”, “violent”, “more violent”, “extremely violent” (continuous controllability). The proposed metric (Eq 3) cannot reflect the said continuous controllability, nor do the showcases in Table 4 support this.

W2: The abstract states that “only a tiny fraction of labeled items are needed”. However, the provided empirical results (shown in Figure 5) are weak. First, the result about how the number of labels available affects the model’s controllability is missing in Figure 5, with only results on disentanglement and completeness being shown there. Second, 0.5+ (achieved via 1% labels) vs 0.9+ seems to be a large gap. It is unclear whether a model with scores of 0.5+ can produce reasonable recommendation lists or has any controllability. Third, there are already plenty of existing works on weakly-supervised disentanglement and are neglected and not compared by the submission.

W3: It is not surprising at all that a supervised method can outperform unsupervised baselines by a large margin. Please make it clear what the challenges are in developing a supervised method for disentanglement.

W4: There is no evidence that the demonstrated controllability is personalized. That is, would the algorithm recommends *different* romantic movies to two users, even if the two users have never seen any romantic movies before, when we ask the algorithm explicitly recommend a romantic movie.

---

### Decision · Program_Chairs · 2021-01-07
**Final Decision**

**Decision:**

Reject

**Comment:**

Although borderline, all reviews are somewhat below the acceptance threshold. The main issue appears to be that the reviewers find the main claims unsupported by the experiments. Other complaints center around the presentation, which could be improved in a revision.